# Conception of Comprehensive Training Program for Family Caregivers: Optimization of Telemedical Skills in Home Care

**DOI:** 10.3390/healthcare13192497

**Published:** 2025-10-01

**Authors:** Kevin-Justin Schwedler, Jan P. Ehlers, Thomas Ostermann, Gregor Hohenberg

**Affiliations:** 1Stabsstelle für Digitalisierung und Wissensmanagement, Hochschule Hamm-Lippstadt, 59063 Hamm, Germany; gregor.hohenberg@hshl.de; 2Fakultät für Gesundheit, Universität Witten-Herdecke, 58455 Witten, Germany; jan.ehlers@uni-wh.de (J.P.E.); thomas.ostermann@uni-wh.de (T.O.)

**Keywords:** telemedicine, family caregivers, home care, digital health, caregiver training, chronic diseases

## Abstract

Background/Objectives: In view of demographic change and the increase in chronic illnesses, home care poses a considerable challenge. Telemedical technologies offer considerable potential for improving the quality of care and relieving the burden on family caregivers. With this study, we aim to develop appropriate training strategies for the use of telemedical applications in home care, focusing on the specific requirements of patients with dementia, heart failure, diabetes mellitus, chronic obstructive pulmonary disease, and stroke. Methods: A comprehensive survey was conducted among 31 family caregivers to record their experience with digital technologies and to analyze caregiver acceptance of these technologies and barriers to their use. The survey comprised 29 questions, including a mix of multiple-choice, Likert scale, and open-ended questions. The internal consistency of the questionnaire was high (Cronbach’s alpha = 0.8876). Results: The results show that although 32% of respondents already use digital technologies, there is a significant need for training and support. Key barriers identified include a lack of technical skills (cited by 45% of respondents), limited access to suitable devices (38%), and privacy concerns (35%). In addition, 90% of respondents expressed a willingness to participate in training programs. Conclusions: Based on the survey results, evidence-based recommendations are provided for the design of training programs tailored to the individual needs of family caregivers. Through a targeted combination of e-learning modules, webinars, and practical exercises, family caregivers can be empowered to take full advantage of telemedical technologies and thus significantly improve the quality of care at home. The results underscore the importance of overcoming technical barriers and providing comprehensive training to ensure the effective use of telemedicine in home care.

## 1. Introduction

In view of the growing number of older people and the increase in chronic illnesses, home care is becoming increasingly challenging [1,2]. Telemedical technologies offer a promising potential solution by improving the quality of care and reducing the burden on caregivers. These technologies include, for example, video calls, telerehabilitation, telemonitoring, mobile health apps and emergency systems that enable continuous monitoring of and care for people in need without requiring them to leave their homes [3]. In this context, the term “telemedicine” covers all medical services that are provided using information and communication technologies, e.g., via videos or apps [4]. Telemedical applications can be used in various areas, such as diagnosis, therapy, and rehabilitation [5]. Telemedicine involves the use of telecommunication technologies as a medium for providing medical information and services to patients remotely. Studies have shown that the use of telemedicine in home care can have a positive impact on various medical conditions, including heart failure, diabetes mellitus, and chronic obstructive pulmonary disease [6]. However, the successful use of telemedicine is heavily dependent on the training and competence of family caregivers, many of whom are currently not sufficiently familiar with these technologies. Only through targeted and needs-based training can relatives exploit the full potential of these technologies and thus optimize home care. It is also important to note that the acceptance of technology depends heavily on how easy it is to use [7].

Current training strategies for caring for relatives in the home environment primarily focus on addressing specific clinical scenarios. However, there is a research gap regarding the development of training strategies that cover the use and benefits of telemedical technologies in home care.

The aim of this study is to address this gap in care by developing and analyzing suitable training concepts for the use of telemedical technologies in home care. The focus is on disease-specific requirements in dementia, heart failure, diabetes mellitus, chronic obstructive pulmonary disease (COPD), and stroke.

This study had on three main objectives:The systematic collection and evaluation of data on the current use of telemedical applications by family caregivers.The identification of relevant barriers that prevent the implementation and effective use of these technologies in the home care context.The analysis of the specific training needs of family caregivers with regard to telemedicine support options.

The following research question is addressed: How can training programs be developed specifically for family caregivers to improve the use of telemedical technologies in home care?

The findings of this study will be incorporated into further development of the research and form the basis for evidence-based recommendations for the development of practice-oriented training programs. These are designed to enable family caregivers to use telemedical technologies competently to increase the quality of home care in the long term.

## 2. Materials and Methods

A combined methodology was used to analyze the training needs of family caregivers and to evaluate the impact of digital technologies on the burden of care and the relationship with the person being cared for. This included a comprehensive analysis of relevant clinical pictures, the selection of suitable telemedical technologies, the development of didactically sound training materials, and the adaptation of training courses to the circumstances of the family caregivers. In addition, a targeted survey was conducted to obtain specific insights and feedback from family caregivers.

Questionnaire development and validation: The questionnaire was developed to gain a comprehensive understanding of the needs, experiences, and challenges of family caregivers regarding the use of digital technologies in home care. The target group was very well known, ensuring the relevance and accuracy of the questions. The questions were carefully selected to cover various aspects of the use and acceptance of digital technologies in home care. The questionnaire consisted of 29 questions, including 12 multiple-choice questions, 7 Likert scale questions, and 10 open-ended questions. The multiple-choice questions were used to collect clear and simple information, the Likert scale questions were used to capture emotions and attitudes, and the open-ended questions were included to obtain individual opinions and detailed insights. The questions were developed based on an extensive literature review and expert feedback to ensure that all relevant topics were covered. Three experts reviewed the questionnaire, and personal interviews were conducted with 10 family caregivers to validate the questions. The internal consistency of the questionnaire was calculated using Cronbach’s alpha, which yielded a value of 0.887595258, indicating high reliability. The questionnaire was further validated by exploratory factor analysis, which identified three factors: (1) the use of telemedicine/digital technologies in the home care of a relative in need of care, (2) acceptance/willingness to undergo training in the use of digital technologies, and (3) demographic data with information on the duration and intensity of care for a relative in need of care. Data analysis was performed using SPSS statistical software (Version 30).

Survey response rate: The questionnaire was distributed to family caregivers who were actively enrolled in a caregiving training course at the time of the survey. Of approximately 45 family caregivers, 31 participated in the survey, representing a response rate of approximately 69%.

Analysis of clinical pictures: First, the central clinical picture was identified for each home care context. Common diseases such as dementia, heart failure, diabetes mellitus, stroke and chronic obstructive pulmonary disease (COPD) were selected with the help of epidemiological data and relevant health reports. These diseases were prioritized due to their high care requirements and the potential applicability of telemedical support.

Selection of telemedical technologies: Specific telemedical technologies were evaluated and selected for each clinical picture. These included telemonitoring devices for monitoring vital signs, video communication systems for regular medical consultations, mobile health applications to support medication adherence and symptom tracking, and emergency systems for rapid response in emergency situations. This selection was supported by a systematic literature review and expert interviews to ensure optimal suitability for the home care sector.

Survey of family caregivers: To supplement the methods mentioned above, a targeted survey was conducted among the family caregivers. The aim was to record their experiences, needs, and challenges in using digital technologies in home care. The survey was completed by 31 people who had attended a care course for family caregivers. The survey results showed that around a third of the respondents already use digital technologies, while the majority are open to using them in the future. The survey also identified key barriers such as technical knowledge, access to suitable devices, privacy concerns, and the complexity and cost of the technologies. These findings were directly applied in the development and adaptation of the training programs.

Combined methodology: The combined methodology included both a systematic literature review and a targeted survey of family caregivers. The literature review provided a comprehensive understanding of the current state of telemedical technologies and training needs, while the survey provided specific insights from the target group. The results from the literature were compared with the survey results to ensure that the training programs were both scientifically sound and relevant to practice.

Development of training materials: Based on the identified training needs, didactic materials were developed, including e-learning modules to teach basic knowledge and practical skills, written instructions and checklists, and expert webinars. The survey results were considered when creating these materials to ensure that they are practical and needs-oriented.

Customization of training courses: A key component of the methodology was adapting the training content to the individual needs and circumstances of family caregivers. Flexible learning formats were developed that can be adapted in terms of time and location to enable easy participation. These include both synchronous and asynchronous learning methods that allow relatives to learn at their own pace and adapt to the situation.

Ethical considerations: All participants were fully informed about the objectives, content, and procedure of the study and provided their written consent to participate voluntarily. The data were collected and evaluated anonymously so that it was not possible to identify individual participants at any time.

Comprehensive approach: This comprehensive methodological approach aimed to ensure the practical relevance and scientific basis of the training programs and to provide family caregivers with the tools and knowledge required to improve the quality of home care through telemedical technologies. The targeted combination of different methods and the direct integration of the survey results ensure that the content is conveyed in a needs-oriented and practical way that meets the specific challenges and needs of family caregivers.

## 3. Results

Research into optimizing care for people in need of care has shown that implementing telemedical systems, particularly in the home sector, has considerable potential [7,8]. To effectively exploit this potential and use the technologies effectively, targeted training for family caregivers is essential. The results of studies on the use of telemedical applications in home care prove their benefits and effectiveness. It is also crucial to analyze the most common and significant clinical pictures of people in need of care to determine which specific technologies are relevant for home care. It is also necessary to adapt the training content to the individual context of these clinical pictures. The results demonstrate how training should be structured to meet the different needs and time constraints of family caregivers to ensure a sustainable improvement in home care. The pedagogical basis of these training courses is also addressed. Based on these considerations, a didactic concept was developed that is tailored to the needs and prior knowledge of the relatives to enable flexible integration of telemedical technologies into the often stressful everyday lives of family caregivers.

### 3.1. Demographic Characteristics of Participants

This survey included a total of 31 family caregivers from different demographics. Table 1 summarizes the most important demographic characteristics of the participants.

The demographic data show a diverse group of family caregivers in terms of age, gender, education level, and caregiving experience. This diversity provides comprehensive insight into the needs of family caregivers and the challenges they face.

### 3.2. Integration of Survey Results

The survey of family caregivers conducted in this study confirmed that around a third of respondents already use digital health technologies such as apps, e-learning modules, webinars, or telemonitoring (see Figure 1 and Figure 2).

The majority of participants have not yet used such technologies; however, they are open to using them in the future (see Figure 3).

In particular, reducing stress improving care quality through using technologies such as telemonitoring and apps were emphasized (see Figure 4). These tools not only enable efficient communication and save time but also contribute to monitoring vital signs and facilitate faster responses to challenges.

However, barriers to use were also identified. A lack of technical knowledge, limited access to suitable devices, data protection concerns, and the complexity and cost of the technologies represent significant barriers (see Figure 5). In addition, many respondents expressed a desire for easier communication with doctors and nurses, centralized management of health data, and support from digital coaches or guides.

These challenges emphasize the importance of offering targeted training and awareness campaigns. The survey participants expressed a clear need for comprehensible e-learning modules, webinars, or classroom training to make it easier to start using digital technologies. The desire for financial support from health insurance companies or employers to facilitate access to these technologies and promote their use was also emphasized.

### 3.3. Major Clinical Pictures

Major medical conditions are those that are not only common but also require complex care involving continuous monitoring, treatment, or support [9]. With the help of telemedical technologies, these care needs can be managed more effectively and efficiently. Dementia, a collective term for a variety of diseases characterized by a progressive loss of cognitive abilities such as memory, thinking, orientation and judgment, is one of the most significant clinical problems. Different types of dementia, with Alzheimer’s being the most common, followed by vascular dementia, Lewy body dementia, and frontotemporal dementia, are particularly common in older people and require intensive care [10]. Due to memory loss, confusion, and the progressive loss of ability, dementia results in a considerable need for care. Specialized care is the only way to ensure the safety and quality of life of those affected [11]. Stroke is a major cause of long-term care dependency and disability [12]. It occurs when the supply of blood to a part of the brain is interrupted or severely reduced. Symptoms include sudden weakness or paralysis on one side of the body, speech and communication problems, and loss of coordination. Rapid recognition and immediate treatment are crucial to minimizing the extent of brain damage. Those affected require continuous care and intensive rehabilitation to improve their functionality and quality of life [13]. Support through telerehabilitation and remote monitoring can help to monitor progress [14].

Heart failure is common in older people and those with cardiovascular disease, which is a chronic condition in which the heart cannot pump enough blood. This condition requires continuous monitoring and adjustment of treatment, including medication and lifestyle changes. Heart failure is characterized by shortness of breath, fatigue, swollen legs, and reduced physical performance. It often occurs together with other chronic diseases, such as high blood pressure or diabetes. Medication intake and lifestyle changes can be supported by telemedical technologies [15].

Diabetes mellitus, a chronic metabolic disease characterized by high blood glucose levels, is common worldwide, especially in older adults. Careful control and monitoring of blood glucose levels are necessary to avoid chronic and acute complications. Regular blood glucose measurements, taking medication such as insulin or oral antidiabetics, dietary management, and physical activity are crucial [16].

Chronic obstructive pulmonary disease (COPD) is one of the most common lung diseases and leads to reduced exercise tolerance. Typical symptoms are chronic coughing, sputum production, and shortness of breath during physical exertion, which significantly impair quality of life and performance. Continuous monitoring and treatment, including medication, oxygen therapy, and inhalation therapy, are required [17].

For all of these conditions, telemedical technologies offer an effective and efficient approach to improving home care by enabling regular monitoring and rapid responses to changes. This ensures high-quality coordination between healthcare professionals, those affected, and their relatives [18].

### 3.4. Importance of Training Courses

For family caregivers to use telemedicine effectively in the home, targeted training courses are extremely important. Various didactic methods are used to impart the necessary knowledge and practical skills for integrating telemedical technologies into everyday care. The survey conducted in this study showed that many family caregivers need training to acquire technical knowledge and learn how to use digital health technologies. The results suggest that training to use digital technologies is an effective tool to support the use of these technologies in caregiving: Respondents with training experience overwhelmingly rated the technologies as helpful, while around 90% of previously untrained individuals would be willing to attend relevant training in principle. These findings underline the importance of targeted training measures to promote the acceptance and effective use of digital applications (see Figure 6 and Figure 7) and the need to design training courses in such a way that they are accessible and helpful for both beginners and advanced users.

The didactic methods can be divided into different categories. First, there are multimedia presentations that familiarize family caregivers with the technical content. A well-known example is PowerPoint presentations, which combine text, images, graphics and videos to present complex information in a clear and appealing way [19]. Furthermore, general videos on the use of telemedical devices or software facilitate understanding through visual explanations [20]. Interactive workshops are another didactic method that can be designed as practical exercises, referred to as hands-on sessions, in which participants test the telemedical devices with the guidance of an expert [21]. The advantage of these practical exercises lies in the direct testing of the devices and applications under expert guidance. Simulations can also be integrated into training courses and offer an opportunity to practice skills in a safe environment through realistic scenarios or role play, also under expert supervision [22]. Online learning platforms offer the opportunity to use e-learning courses that ensure flexible and continuous learning. E-learning courses can be integrated into the everyday lives of family caregivers so that they can learn at their own pace. It is important that the e-learning content is structured well and laid out clearly to correctly orient the relatives on the online platform [23]. It is particularly noteworthy that 47.6% of the respondents prioritized e-learning as their preferred form of training, which indicates a pronounced need for learning opportunities that are flexible in terms of time and location to acquire digital skills (see Figure 8).

Live online webinars allow participants to have their questions answered directly and to exchange ideas with other carers [24]. In addition to practical presentations and knowledge transfer through videos, written materials can be an important didactic tool that can be integrated into training courses. Manuals or guides are often used to provide detailed documentation, such as step-by-step instructions. FAQs, which provide answers to frequently asked questions, can serve as a quick references [25]. Mobile apps offer an opportunity to interactively involve family caregivers in the learning process, for example, through learning modules or quizzes, to test and deepen the knowledge imparted. Mobile apps can also remind users of training dates, exercises, or tasks so that they can follow the course regularly [26].

When implementing didactic measures, it is crucial to respond to the needs of family caregivers and to adapt and design the training courses accordingly. Three key aspects must be taken into account: flexibility, individualization, and easy access [27]. These adaptations ensure that the training courses take into account the diverse life situations, previous knowledge, and technical skills of the family caregivers. Flexibility is key, and it is important to ensure that different learning formats are available. A wide range of formats can accommodate family members’ different preferences and schedules. These formats include learning apps, webinars, online modules, and face-to-face events. Time flexibility plays a key role, as caregiving often involves unpredictable and stressful schedules. Therefore, both synchronous and asynchronous learning methods should be available to facilitate an individual learning pace [28]. A modular content structure also supports clear and organized training. Given the unique challenges that family caregivers face in their home environment, customized training is essential. Different medical conditions require specific applications of telemedical technologies, which is why training should be adapted to specific care needs. The different levels of experience of relatives must also be taken into account: simple and understandable instructions are important for less experienced carers, while more in-depth content should be provided for advanced users. A learning management system can be used to create individual learning paths so that relatives’ progress can be tracked and specific needs can be recognized and considered. It is also important to clearly define the target group for training, as family caregivers often face significant challenges and need easy access to support. Clearly worded and understandable materials are essential, and visual aids and step-by-step guides can be considered helpful. In addition, easily accessible support options, such as hotlines, self-help groups, or counseling services, should be made available to provide specialist help when needed. Through these measures, training can be better adapted to the real needs and conditions of family caregivers, which increases the likelihood that training services will be accepted and used effectively, leading to the successful use of telemedical technologies in home care [29].

The survey results also highlighted the need for additional support services such as telephone advice or technical hotlines to counter uncertainties when using digital technologies in everyday home care (see Figure 9).

### 3.5. Connection Technology and Clinical Picture

The integration of telemedical innovations into home care can make everyday life easier for people in need of care and their families, as these technologies offer individual solutions for various illnesses.

These technologies enable continuous monitoring, timely interventions and comprehensive support, which contribute to a significant improvement in the quality of care. The respondents believe that in general, digital technologies have the potential to improve the quality of care in the home environment, particularly with respect to continuous monitoring and timely responses to change (see Figure 10).

When evaluating which technologies are suitable for certain clinical pictures, these tools are clearly making things easier. With regard to dementia, telemonitoring apps, alarm systems, and GPS tracking can support the care of affected individuals at home by ensuring their safety, monitoring their whereabouts and raising an alarm if they leave a defined area. Telemonitoring apps are able to detect changes in everyday life, as they highlight behavioral patterns that could indicate deterioration or special needs, thereby increasing the safety of those in need of care [30]. Telemonitoring devices are also extremely useful in caring for individuals with heart failure, as they enable continuous monitoring of blood pressure, heart rate, and weight and transmit the data directly to medical staff. Health apps can also remind patients to take their medication and document symptoms, facilitating early detection of deterioration and rapid adjustment of therapy. Relatives benefit as the health of those in need of care is regularly monitored [31]. Furthermore, these applications are also useful for patients with diabetes mellitus, as they can help track and analyze blood glucose measurements, insulin doses, diet, and physical activity. They improve blood glucose control by providing real-time data that support disease management through structured data collection and reminders. In the case of chronic obstructive pulmonary disease (COPD), telerehabilitation can support home care via physiotherapy exercises and breathing techniques that are guided and monitored online. This promotes rehabilitation measures in the home environment and has a positive effect on physical performance and quality of life. Telerehabilitation is also applicable for stroke patients, for whom motor and cognitive exercises can be provided online and progress monitored. Video communication plays a crucial role in enabling regular reviews of and adjustments to the rehabilitation plan by doctors and therapists. This ensures ongoing rehabilitation in the home environment with professional guidance. Virtual checks using video communication also support relatives in the care and motivation of stroke patients in the home context [32]. The advantage of digital technologies in home care is made clear by the results of the survey: the possibility of location-independent access, the constant availability of digital services, and the use of real-time data to monitor health status were particularly emphasized. These aspects enable a faster response in acute situations, better accessibility of specialists, and greater security of care for both people in need of care and their carers (see Figure 11).

### 3.6. Connection: Didactic Tools and Technologies

In addition to identifying suitable technologies for specific medical conditions, it is also important to investigate how educational training measures can facilitate access to technology for family caregivers [33]. The survey showed that a large proportion of the family caregivers surveyed would like training courses to enable them to use telemedical applications safely and efficiently (cf. Figure 7). Demonstrative or explanatory videos are an excellent way of explaining how to use telemonitoring devices and mobile health applications, for example. Such videos can demonstrate in detail how to use the respective applications and devices. In addition, software can be used to create interactive learning modules to simulate the use of telemedical applications and devices, giving trainees practical insight into the use of these technologies. A particular advantage of these videos is that they can be viewed asynchronously and repeatedly. In addition to explanatory videos, which are known as e-learning videos, live demonstrations and simulation tools offer the opportunity to experience the technologies in practical use. Telerehabilitation platforms or emergency alarm systems can therefore be demonstrated in real time. Building on this, practical exercises with the devices can be carried out under the guidance of a trainer in a controlled environment. Simulations are used to recreate real care situations and learn how to use the technologies. Online learning platforms enable the creation of specific modules that offer relatives a structured online course teaching them to use these technologies step by step. The respondents considered time-flexible and modular online courses to be particularly suitable for combining training with everyday care work (see Figure 8). Interactive webinars offer the opportunity to impart expert knowledge on telemedicine in home care through live online seminars. In addition, they make direct question-and-answer situations possible, in which the concerns or questions of relatives can be addressed. Live webinars also offer family caregivers the opportunity to exchange information with each other. In moderated group discussions, participants can share experiences and best practices so that newcomers can be supported by more experienced carers. Written materials provide an additional training opportunity in relation to telemedical technologies. Manuals or guides often contain written instructions that provide step-by-step guidance [34].

### 3.7. General Overview: Effective Training for the Use of Telemedical Technologies in Home Care

In order to best support family caregivers in their daily lives, it is essential to offer training that is tailored to specific medical conditions and the corresponding telemedical technologies. Asynchronous, online-based training methods are particularly beneficial as they are flexible and easily accessible. These trainings should include clear recommendations for different medical conditions.

Dementia: GPS tracking, alarm systems, and telemonitoring apps are useful tools for caring for people with dementia. Online courses and webinars explaining how to use these devices are particularly suitable. E-learning videos illustrate specific scenarios, and interactive tutorials support learning how to use the technology. These training materials are available at any time via online platforms. Technical support should also be available around the clock [35].

Heart failure: Telemonitoring devices and mobile health applications are highly relevant for the care of patients with heart failure. Online modules with step-by-step instructions, regular webinars, and interactive simulation programs are suitable training methods. They can be accessed via a central online platform that enables relatives to learn at their own pace and receive support from experts.

Diabetes mellitus: Applications are of great importance in the care of diabetes patients. Online learning courses that explain how to use these applications, e-learning videos, and tutorials are suitable training methods.

### 3.8. Comparison of This National Study with International Studies

The study presented in this article examines the training needs and use of telemedical technologies by family caregivers in Germany. In order to validate and expand on the findings of this study, relevant international studies and systematic reviews are presented and compared below. These international sources can be divided into three main categories: studies on the general relevance and introduction of telemedical training, studies on the training of family caregivers, and studies on the benefits of telemedicine in home care.

General relevance and introduction of telemedical training

These studies show that telemedical training is relevant for healthcare professionals and how it can be introduced. Although they focus primarily on healthcare professionals, the findings can also be applied to family caregivers. For example, these studies emphasize the need for targeted training programs to improve the acceptance and use of telemedical technologies. Yaghobian et al. [36] examined the implementation of telemedical training at medical schools in France and found that telemedical training is limited, with a strong need for expansion. Pourmand et al. [37] emphasize that teaching telemedicine to medical students is essential to best prepare the next generation of healthcare providers. Stovel et al. [38] recommend centering curricula on a competency-based, outcome-oriented framework such as CanMEDS to ensure comprehensive and effective telemedical education.

2.Training courses for family caregivers

These studies specifically examine training needs and methods for family caregivers. They show that targeted training programs can reduce the burden on family caregivers and improve the quality of care. Sitges-Maciá et al. [29] conducted a narrative review of the effects of e-health training and social support interventions for informal caregivers of people with dementia and found that supportive interventions through telemedicine offer caregivers the opportunity to learn necessary skills and maintain social networks. Van Houtven et al. [39] evaluated the effectiveness of a competency training program for caregivers of people with functional or cognitive impairments and found that the program led to a sustained improvement in the experiences of caregivers and patients with VA care.

3.Benefits of telemedicine in home care

These studies examine the general benefits of telemedicine in home care and how it can help family caregivers. They show that telemedicine improves the quality of care and reduces the burden on family members by enabling continuous monitoring and rapid responses to changes. Chi and Demiris [40] conducted a systematic review of studies using telemedical interventions to support family caregivers and found that telemedicine has positive effects on the treatment of chronically ill patients and on home and hospice care. Graven et al. [41] analyzed the components and outcomes of telemedical interventions for family members of people with chronic illnesses and found that telemedicine is an effective tool for supporting caregivers and leads to a significant improvement in treatment outcomes. Yang et al. [42] evaluated the effects of telemedicine on the stress, anxiety, depression, and quality of life of informal caregivers of patients in palliative care and found that telemedicine can alleviate the burden of care and anxieties of family caregivers but does not significantly reduce depression or improve quality of life. Ye et al. [43] examined implementation strategies for providing telemedicine to support dementia patients and their caregivers and found that telemedicine can alleviate care-related problems for dementia patients and their family caregivers. Lai et al. [44] found that telemedicine via video conferencing improves the resilience and well-being of both people with noncommunicable diseases and their caregivers at home. Amiri et al. [45] examined the goals, outcomes, facilitators, and barriers to the use of telemedical systems for Alzheimer’s patients and their caregivers and found that telemedical systems can be implemented for a variety of reasons and offer numerous benefits, although they also present challenges.

This national study shows that one-third of family caregivers already use digital technologies, and the majority are open to using them in the future. This is consistent with the findings of international studies, which also identify acceptance and willingness to use telemedical technologies. The barriers identified in this study, such as a lack of technical knowledge and data protection concerns, are also present in international studies. These studies emphasize the need for targeted training programs to overcome these barriers. This national study reveals specific characteristics that cannot necessarily be generalized at the international level. For example, data protection concerns and technical challenges are particularly pronounced in Germany, which can be attributed to strict data protection laws and differences in technical infrastructure. These factors must be taken into account when developing training programs and implementing telemedical technologies. In contrast, the international studies show that the basic needs and challenges of family caregivers are similar worldwide, indicating the transferability of certain training approaches and best practices.

## 4. Discussion

Interpretation of results: The implementation of telemedical systems in home care offers considerable potential for improving care for people in need of care. The survey conducted in this study shows that around a third of family caregivers already use digital technologies such as apps, e-learning modules, webinars, or telemonitoring and report reduced stress and improved care. The majority of relatives are open to using digital technologies in the future, emphasizing the need for effective training and information services. The survey clearly identified key barriers such as technical knowledge, access to suitable devices, privacy concerns, and the complexity and cost of the technologies. The participants expressed a need for comprehensible e-learning modules, webinars, and financial support from health insurance companies or employers. Flexible and modular training content is crucial to accommodating the different life situations and schedules of caregiving family members.

Practical implications: Telemedical technologies offer an effective solution for managing the care needs of individuals with common conditions such as dementia, heart failure, diabetes mellitus, chronic obstructive pulmonary disease, and stroke. Continuous monitoring, rapid responses to changing conditions, and the promotion of communication between family caregivers and healthcare professionals are of central importance in these contexts. The research question of this study was, “How can training programs be developed specifically for family caregivers to improve the use of telemedical technologies in home care?” The findings suggest that targeted training programs should include e-learning modules, webinars, and practical exercises tailored to the specific needs of family caregivers. Addressing barriers such as a lack of technical knowledge and privacy concerns, along with financial support, can enhance the integration of these technologies into daily care routines.

Study limitations: This study has several limitations that should be acknowledged. First, its exploratory nature means that the findings are preliminary and require further validation. Second, the reliance on self-reported data may have introduced bias, as the participants may have overestimated or underestimated their experiences and needs. Third, the small sample size of 31 participants limits the generalizability of the results. Finally, the lack of representativeness means that the findings may not be applicable to all family caregivers, particularly those from different cultural or socio-economic backgrounds.

## 5. Conclusions

In summary, this work shows that targeted training, technical support, and transparent data protection guidelines are essential to promote the acceptance and use of telemedical technologies. The survey conducted in this study contributed significantly to identifying the needs of family caregivers and the obstacles they face and developing concrete recommendations for action. By taking these results into account, home care can be improved in the long term. The main findings of this study indicate a clear need for targeted training programs that address the specific requirements of family caregivers handling chronic conditions such as dementia, heart failure, diabetes mellitus, COPD, and stroke. Practical implications include the development of e-learning modules, webinars, and practical exercises tailored to the needs of family caregivers. Financial support from health insurance companies or employers could facilitate access to these technologies and promote their use. Future research should aim to address these limitations by including larger and more diverse samples, as well as employing objective measures of technology use and family caregiver burden. This study contributes to the growing body of literature on telemedicine in home care by providing evidence-based recommendations for the development of practice-oriented training programs. The results of this national study are largely comparable to those of international studies: all show that targeted training programs and the integration of telemedical technologies can improve the quality of care and reduce the burden on family caregivers. The international studies offer additional insights and best practices that can be incorporated into the development and implementation of training programs in Germany. However, it is important to take specific national characteristics into account in order to make the training programs effective and relevant.

## Figures and Tables

**Figure 1 healthcare-13-02497-f001:**
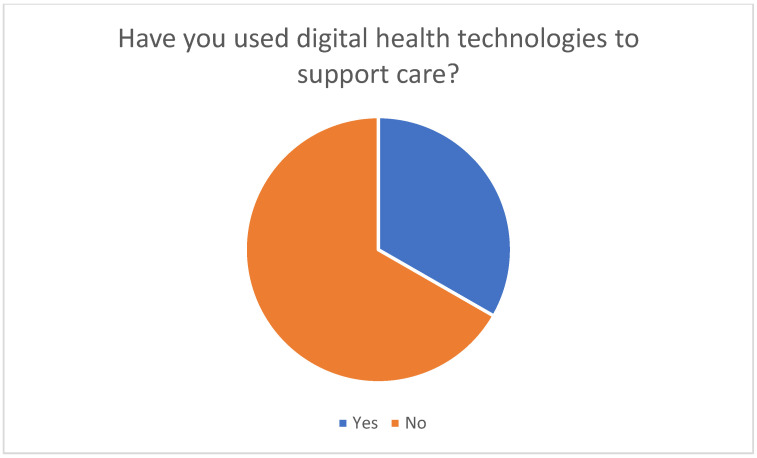
Previous use of digital health technologies to support nursing activities.

**Figure 2 healthcare-13-02497-f002:**
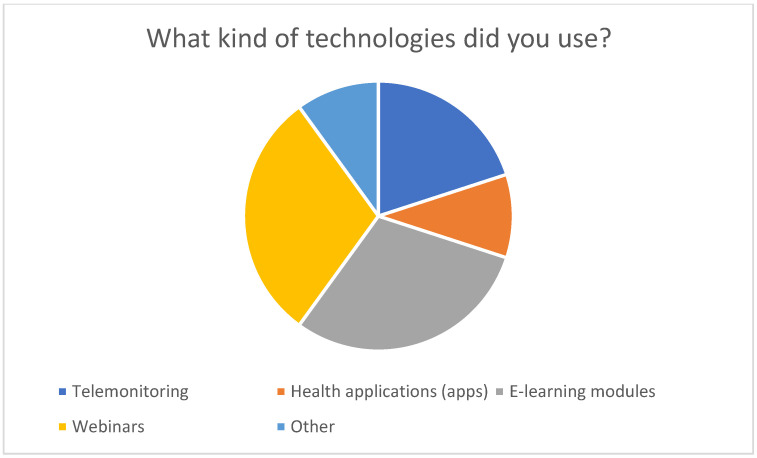
Types of technologies used.

**Figure 3 healthcare-13-02497-f003:**
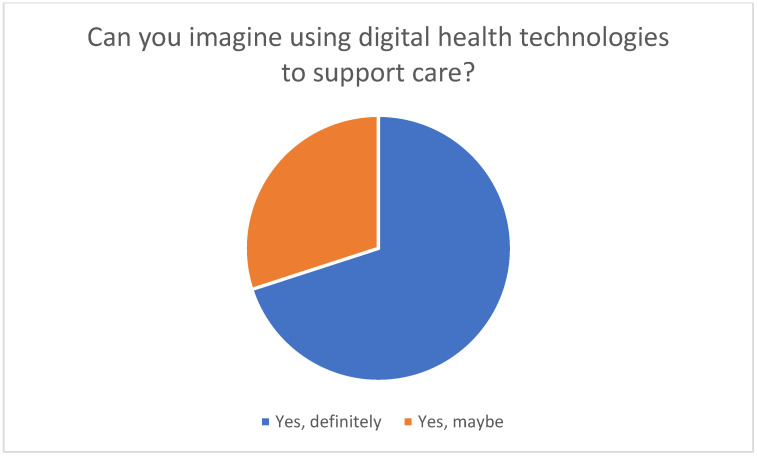
Conceivability of using digital health technologies to support nursing care.

**Figure 4 healthcare-13-02497-f004:**
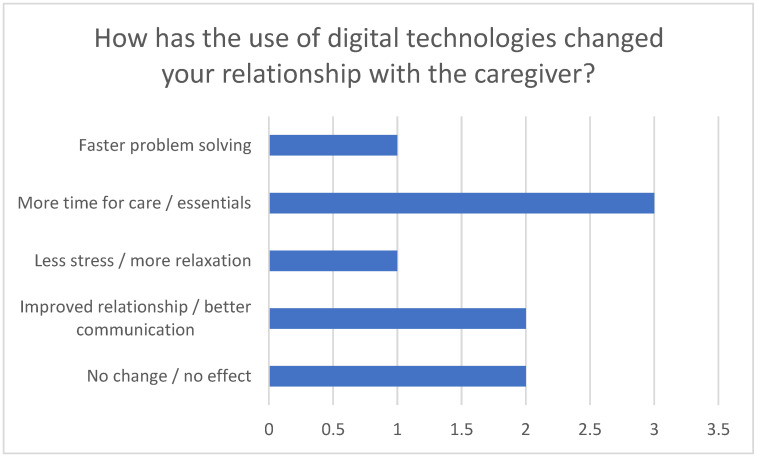
Change in the relationship with the person requiring care through the use of digital technologies.

**Figure 5 healthcare-13-02497-f005:**
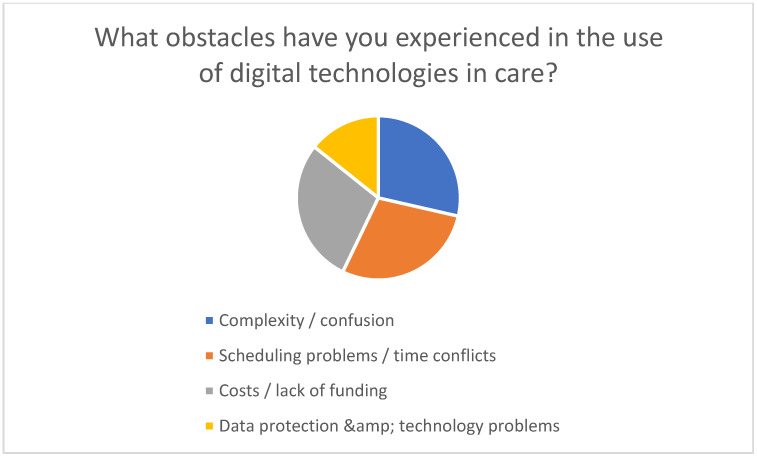
Challenges encountered when using digital technologies in nursing care.

**Figure 6 healthcare-13-02497-f006:**
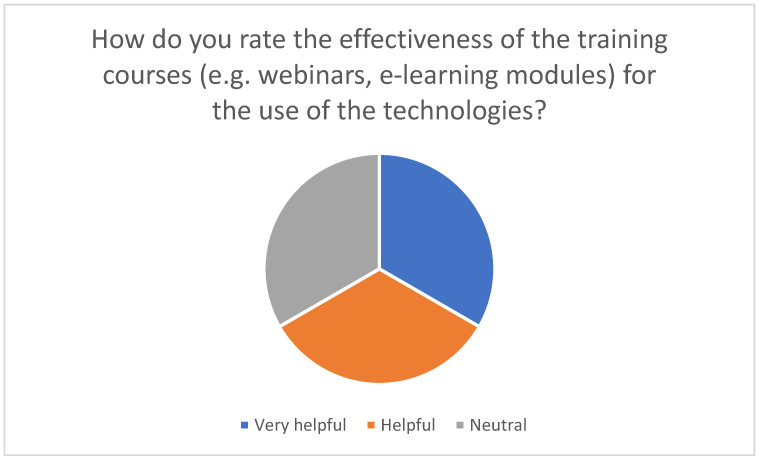
Assessing the effectiveness of training programs (e.g., webinars, e-learning modules) on the use of technology.

**Figure 7 healthcare-13-02497-f007:**
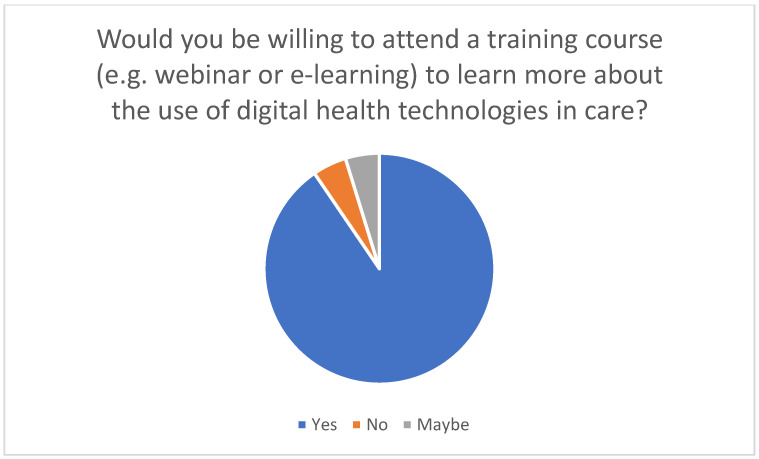
Willingness to participate in training (e.g., webinar or e-learning) to learn more about the use of digital health technologies in care.

**Figure 8 healthcare-13-02497-f008:**
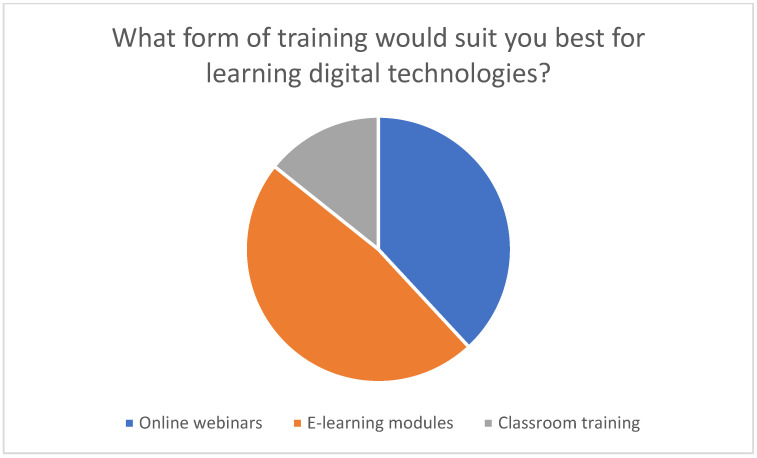
Preferred training course for learning digital technologies.

**Figure 9 healthcare-13-02497-f009:**
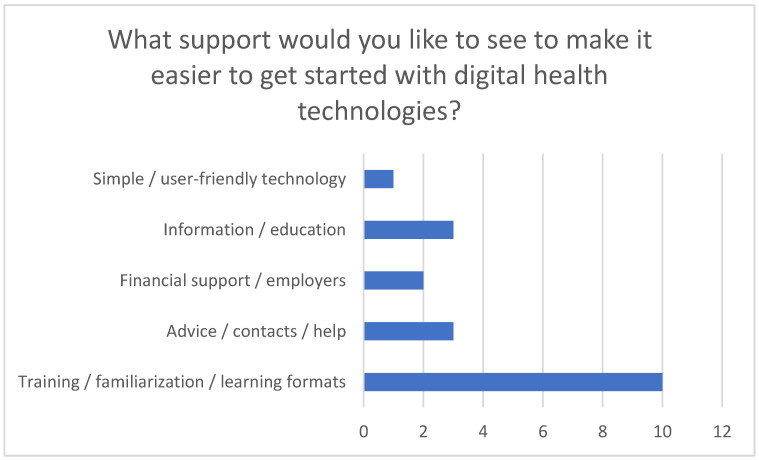
Support services needed to start using digital health technologies.

**Figure 10 healthcare-13-02497-f010:**
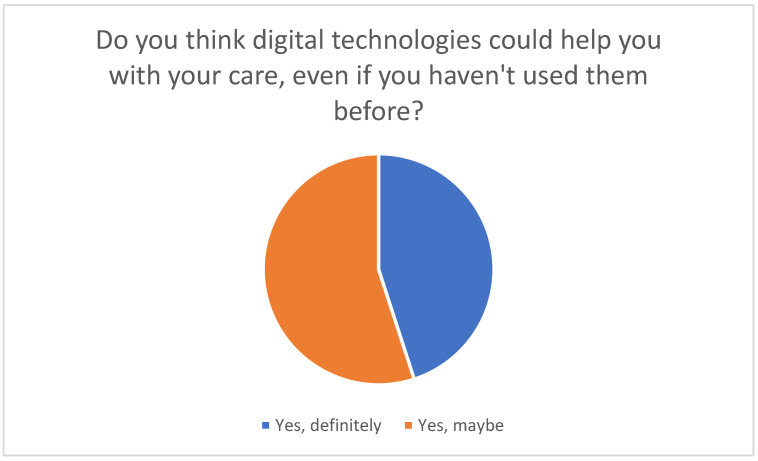
Opinions on the potential benefits of digital technologies for caregiving, even without previous use.

**Figure 11 healthcare-13-02497-f011:**
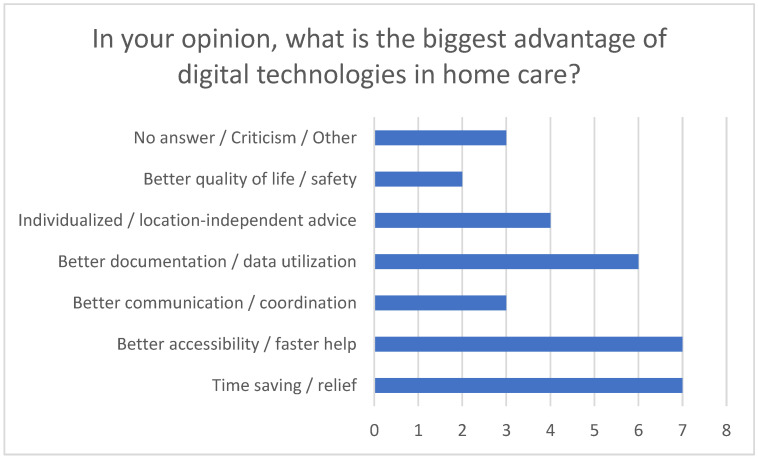
The greatest advantages of digital technologies in home-based care from the perspective of the respondents.

**Table 1 healthcare-13-02497-t001:** Demographic characteristics and caregiving experience of survey participants.

Person	How Old Are You?	What Is Your Gender?	What Is Your Highest Level of Education?	How Long Have You Been Caring for a Loved One?	How Many Hours Per Week Do You Spend on Average Providing Care?
1	40–49	Female	Secondary school diploma	1–3 years	21–30 h
2	Under 30	Female	University degree	Less than 6 months	More than 30 h
3	60–69	Female	Secondary school diploma	More than 3 years	21–30 h
4	50–59	Female	Vocational training	More than 3 years	21–30 h
5	Under 30	Female	Vocational training	More than 3 years	10–20 h
6	Under 30	Male	Secondary school diploma	Less than 6 months	Less than 10 h
7	50–59	Female	University degree	More than 3 years	10–20 h
8	40–49	Male	Vocational training	More than 3 years	21–30 h
9	Under 30	Female	Vocational training	6 months–1 year	Less than 10 h
10	50–59	Female	University degree	6 months–1 year	Less than 10 h
11	50–59	Male	Secondary school diploma	Less than 6 months	More than 30 h
12	50–59	Female	University degree	1–3 years	10–20 h
13	40–49	Female	Vocational training	More than 3 years	21–30 h
14	30–39	Female	Secondary school diploma	More than 3 years	10–20 h
15	40–49	Female	Vocational training	Less than 6 months	Less than 10 h
16	60–69	Female	Vocational training	1–3 years	10–20 h
17	40–49	Female	Vocational training	1–3 years	10–20 h
18	50–59	Female	High school diploma	More than 3 years	10–20 h
19	Under 30	Male	University degree	Less than 6 months	Less than 10 h
20	Under 30	Male	Vocational training	More than 3 years	10–20 h
21	Under 30	Female	High school diploma	Less than 6 months	10–20 h
22	Under 30	Female	Vocational training	More than 3 years	10–20 h
23	40–49	Female	Vocational training	More than 3 years	More than 30 h
24	30–39	Female	High school diploma	Less than 6 months	Less than 10 h
25	40–49	Male	Other	More than 3 years	10–20 h
26	30–39	Female	Junior high school diploma	1–3 years	21–30 h
27	Under 30	Female	University degree	Less than 6 months	Less than 10 h
28	40–49	Diverse	Vocational training	6 months–1 year	10–20 h
29	under 30	Female	High school diploma	Less than 6 months	More than 30 h
30	30–39	Female	Junior high school diploma	Less than 6 months	21–30 h
31	40–49	Female	High school diploma	More than 3 years	10–20 h

## Data Availability

The data are contained within the article.

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
