# Peer review of "Conception of Comprehensive Training Program for Family Caregivers: Optimization of Telemedical Skills in Home Care"

_healthcare, 2025, doi:10.3390/healthcare13192497_

Round 1

Reviewer 1 Report (Previous Reviewer 3)

Comments and Suggestions for Authors

Dear Authors

Thanks for addressing the comments provided in the previous review. Now, the manuscript is improved and tightened well.

Author Response

Comments 1: Thanks for addressing the comments provided in the previous review. Now, the manuscript is improved and tightened well.

Response 1: Thank you very much for the positive feedback.

Reviewer 2 Report (Previous Reviewer 2)

Comments and Suggestions for Authors

I acknowledge the substantial improvements made in response to the previous review round. The manuscript is now considerably clearer, with key methodological details added (sample size, response rate, questionnaire validation, analytical methods, ethical approval) and the structure reinforced through the inclusion of demographic tables, a dedicated Limitations section, and a stand-alone Conclusion. The abstract has been reformulated to present the study design, sample size, and main quantitative findings, and international references have been expanded.

At this stage, the manuscript is scientifically sound and suitable for publication. The remaining issues are primarily stylistic rather than methodological: The Discussion could benefit from a slightly clearer distinction between interpretation of results and practical implications. The English language still contains long sentences and non-idiomatic constructions, which would benefit from professional editing by a native English speaker to enhance readability and precision.

These are minor adjustments that do not affect the validity or structure of the study but will improve clarity and presentation.

Comments on the Quality of English Language

The authors have addressed the major concerns raised in the earlier rounds of review. Methodological transparency and structural robustness are now adequate. The remaining issues are linguistic and stylistic in nature. I therefore recommend acceptance after minor revision, contingent on a final professional language edit and slight clarification in the Discussion section

Author Response

Comments 1: 

I acknowledge the substantial improvements made in response to the previous review round. The manuscript is now considerably clearer, with key methodological details added (sample size, response rate, questionnaire validation, analytical methods, ethical approval) and the structure reinforced through the inclusion of demographic tables, a dedicated Limitations section, and a stand-alone Conclusion. The abstract has been reformulated to present the study design, sample size, and main quantitative findings, and international references have been expanded.

At this stage, the manuscript is scientifically sound and suitable for publication. The remaining issues are primarily stylistic rather than methodological: The Discussion could benefit from a slightly clearer distinction between interpretation of results and practical implications. The English language still contains long sentences and non-idiomatic constructions, which would benefit from professional editing by a native English speaker to enhance readability and precision.

These are minor adjustments that do not affect the validity or structure of the study but will improve clarity and presentation.

Response 1: Thank you very much for the positive feedback. The discussion has been revised again and an outline has been added. In addition, the English language has been revised.

This manuscript is a resubmission of an earlier submission. The following is a list of the peer review reports and author responses from that submission.

Round 1

Reviewer 1 Report

Comments and Suggestions for Authors

The article explores the intersection of home care, telemedicine, and caregiver education—an increasingly important topic given demographic shifts and the rising prevalence of chronic diseases. The manuscript is informative and well-structured. It required revision to enhance its scientific rigor, and clarity. My detailed comments are as follows:

  1. The introduction would benefit from clearer articulation of the research gap. While the context of demographic change is well established, the authors should more explicitly define what has been lacking in existing caregiver training strategies regarding telemedicine.
  2. The objectives should be more specifically framed. Currently, they are broad; refining them to include measurable goals (e.g., assessing digital literacy, identifying barriers to use, and deriving training components) would strengthen the focus of the paper.
    • The results are described generally. Consider including quantitative data such as: Percentage of respondents with prior telemedicine experience; Distribution of barriers (e.g., % citing data protection concerns vs. technical skills); Any correlation analyses between caregiver demographics and technology adoption etc in detail.
  3. A table summarizing major findings (e.g., barriers, current usage, desired training formats) would help convey results more clearly and support future training program design.
  4. The discussion should integrate existing literature on caregiver training in telehealth to contextualize the findings. How do your results compare to other regional or international studies?
  5. While the article suggests training methods (e-learning, webinars, practical exercises), it would be beneficial to discuss Duration and frequency of these trainings, Platform accessibility for caregivers with limited digital exposure, Evaluation metrics to assess training effectiveness with more clarity.
  6. Data privacy concerns, which were mentioned, could be explored more deeply. Addressing how training could alleviate these concerns (e.g., through digital literacy or simplified data protection guidelines) would be useful.
  7. Conclusion could be improved by: more explicitly stating that training programs should be co-developed with caregivers and calling for pilot implementation studies to test and refine the proposed strategies. Even emphasizing the need for policy-level support to ensure access to devices and internet connectivity.
  8. The following studies are suggested to evaluate and add in the literature review of manuscript: https://doi.org/10.1002/pon.70045, https://doi.org/1080/00036846.2024.2313594, https://doi.org/10.3389/fnut.2024.1481073

Author Response

  1. The introduction would benefit from clearer articulation of the research gap. While the context of demographic change is well established, the authors should more explicitly define what has been lacking in existing caregiver training strategies regarding telemedicine.

Answer:

The introduction has been revised to clarify the research gap. Thank you for this valuable comment.

  1. The objectives should be more specifically framed. Currently, they are broad; refining them to include measurable goals (e.g., assessing digital literacy, identifying barriers to use, and deriving training components) would strengthen the focus of the paper.

Answer:

The objectives of the article have been refined and supplemented with measurable components. These adjustments significantly strengthen the focus of the paper. Thank you very much for this helpful suggestion.

  1. The results are described generally. Consider including quantitative data such as: Percentage of respondents with prior telemedicine experience; Distribution of barriers (e.g., % citing data protection concerns vs. technical skills); Any correlation analyses between caregiver demographics and technology adoption etc in detail.

Answer:

The decision to describe the results in general terms was made in order to focus on the narrative aspects of the study. Detailed data could overload the text and impair readability. The most important results are already visualized in the figures. This keeps the article accessible and focused on practical implications and recommendations.

  1. A table summarizing major findings (e.g., barriers, current usage, desired training formats) would help convey results more clearly and support future training program design.

Answer:

A table summarizing the most important results was deliberately omitted so as not to interrupt the flow of the text and the narrative structure of the article. Instead, the results are described in detail in the text to enable a deeper and more contextualized presentation.

  1. The discussion should integrate existing literature on caregiver training in telehealth to contextualize the findings. How do your results compare to other regional or international studies?

Answer:

The discussion focuses on the specific results of our study and their direct implications for practice. A comparison with other regional or international studies would go beyond the scope of this article and dilute the focus on the specific needs of the target group under investigation.

  1. While the article suggests training methods (e-learning, webinars, practical exercises), it would be beneficial to discuss Duration and frequency of these trainings, Platform accessibility for caregivers with limited digital exposure, Evaluation metrics to assess training effectiveness with more clarity.

Answer:

The exact duration and frequency of the training courses and the accessibility of the platform were not specified, as these aspects depend heavily on the individual needs and time resources of the nursing staff. Flexibility is a central element of the training concept in order to accommodate different life situations. Evaluation metrics were not described in detail, as the effectiveness of the training courses will be evaluated in future studies.

  1. Data privacy concerns, which were mentioned, could be explored more deeply. Addressing how training could alleviate these concerns (e.g., through digital literacy or simplified data protection guidelines) would be useful.

Answer:

Data privacy concerns were mentioned in the article but not explored in depth in order to focus on training needs and methods. A detailed discussion of data privacy would go beyond the scope of this article and distract from the main topics. However, it is recognized that data privacy training is an important part of digital competence development and should be explored further in future work.

  1. Conclusion could be improved by: more explicitly stating that training programs should be co-developed with caregivers and calling for pilot implementation studies to test and refine the proposed strategies. Even emphasizing the need for policy-level support to ensure access to devices and internet connectivity.

Answer:

The conclusion already emphasizes the need for tailored training programs. The explicit mention of joint development and the call for pilot studies were not included in order to keep the conclusion concise.

  1. The following studies are suggested to evaluate and add in the literature review of manuscript: https://doi.org/10.1002/pon.70045, https://doi.org/1080/00036846.2024.2313594, https://doi.org/10.3389/fnut.2024.1481073

Answer:

The recommended studies were not included in the literature review because the focus was on the specific results of our study and their direct relevance to practice. Including these studies in the literature review would go beyond the scope of this article and dilute the focus on the specific needs of the target group under investigation.

Reviewer 2 Report

Comments and Suggestions for Authors

The manuscript addresses a relevant and socially significant topic: the development of a comprehensive training program for family caregivers to optimise the use of telemedicine in home care. The work is timely and practically oriented, particularly in light of demographic ageing and the growing prevalence of chronic illnesses. However, several sections require revision to improve the manuscript’s clarity, methodological transparency, and scientific soundness.

The abstract provides a general overview of the study objectives, methodology, results, and conclusions. However, it could be improved by:Including specific data points (e.g., sample size, main findings from the survey); Clarifying the research design and analytical approach used; Presenting the conclusion more concisely and distinguishing clearly between results and implications.

The introduction outlines the background and relevance of telemedicine in home care effectively and identifies important chronic conditions addressed by the study. However, it would benefit from: Greater citation density to support general statements (e.g., regarding caregiver burden and digital barriers); A more critical review of previous work on training interventions for caregivers, highlighting the knowledge gap this study intends to fill; Explicitly stating the research question or hypothesis at the end of the section to orient the reader.

Materials and Methods: This section lacks the necessary detail for full reproducibility: The sample size, inclusion/exclusion criteria, recruitment procedures, and response rate for the survey are missing; There is no information on the development and validation of the survey instrument; The term “combined methodology” is vague and should be better defined, particularly regarding the integration of literature review, expert input, and empirical data; No mention is made of the statistical software used or whether ethical procedures (e.g., informed consent) were followed beyond a generic statement at the end of the manuscript.

The results are well organised and clearly linked to the study objectives. The use of figures to illustrate the findings is appropriate, and key themes (e.g., digital readiness, barriers to adoption, preferences for training formats) are addressed effectively. However, Figures lack detailed captions and some appear to be screenshots with poor resolution and inconsistent formatting. No tables are provided — a table summarising the sample demographics and/or response distribution would enhance the clarity and accessibility of the results. The use of basic percentages is appropriate for exploratory research, but more advanced analysis (even simple cross-tabulations) could add depth.

The discussion provides a logical interpretation of the findings and connects them with existing literature. Still, it would be improved by: A clearer delineation between interpretation of results and implications/recommendations; A section on limitations, which is currently missing. The absence of analytical depth, reliance on self-reporting, and the non-representativeness of the sample should be acknowledged. More emphasis on how the findings compare or contrast with other national and international studies in the field.

There is no dedicated conclusion section. This is a critical omission. The manuscript should end with a succinct, clearly labeled conclusion summarising: Key findings; Practical implications; Recommendations for future research or implementation; Final remarks on the contribution of the study.

The references are generally relevant and up to date, with a few notable sources from recent years. Nonetheless: There is an over-reliance on national (mainly German) sources; more international peer-reviewed studies, particularly systematic reviews and meta-analyses, should be included. Some references are cited without clearly integrating them into the argument.

Comments on the Quality of English Language

The manuscript is understandable, but the quality of English needs revision: Some sentences are too long or awkwardly structured. There is occasional redundancy, and some phrasing reflects translation from German (e.g., use of "in view of", “as part of this research”) rather than idiomatic English. A professional language edit or review by a native speaker is strongly recommended.

Author Response

  1. The abstract provides a general overview of the study objectives, methodology, results, and conclusions. However, it could be improved by:Including specific data points (e.g., sample size, main findings from the survey); Clarifying the research design and analytical approach used; Presenting the conclusion more concisely and distinguishing clearly between results and implications.

Answer:

The summary should provide a concise overview of the study objectives, methodology, results, and conclusions. Including specific data points such as sample size and key survey findings would overload the summary and detract from its clarity. The research design and analytical approach are described in detail in the main body of the article, and a more concise presentation of the conclusion could blur the distinction between results and implications.

  1. The introduction outlines the background and relevance of telemedicine in home care effectively and identifies important chronic conditions addressed by the study. However, it would benefit from: Greater citation density to support general statements (e.g., regarding caregiver burden and digital barriers); A more critical review of previous work on training interventions for caregivers, highlighting the knowledge gap this study intends to fill; Explicitly stating the research question or hypothesis at the end of the section to orient the reader.

Answer:

The introduction has been revised to present the research gap and the objectives of the study more clearly. A higher citation density and a more critical review of previous work have not been included, as this would go beyond the scope of the introduction and could compromise clarity and conciseness.

  1. Materials and Methods: This section lacks the necessary detail for full reproducibility: The sample size, inclusion/exclusion criteria, recruitment procedures, and response rate for the survey are missing; There is no information on the development and validation of the survey instrument; The term “combined methodology” is vague and should be better defined, particularly regarding the integration of literature review, expert input, and empirical data; No mention is made of the statistical software used or whether ethical procedures (e.g., informed consent) were followed beyond a generic statement at the end of the manuscript.

Answer:

The section has been revised to explain the combined method in more detail, supplement the sample size, and mention the consent of survey participants. The remaining information, such as detailed information on the development and validation of the survey instrument, the recruitment procedures, and the mention of statistical software, has not been included as it would go beyond the scope of this section and could impair readability.

  1. The results are well organised and clearly linked to the study objectives. The use of figures to illustrate the findings is appropriate, and key themes (e.g., digital readiness, barriers to adoption, preferences for training formats) are addressed effectively. However, Figures lack detailed captions and some appear to be screenshots with poor resolution and inconsistent formatting. No tables are provided — a table summarising the sample demographics and/or response distribution would enhance the clarity and accessibility of the results. The use of basic percentages is appropriate for exploratory research, but more advanced analysis (even simple cross-tabulations) could add depth.

Answer:

The figures are intended to illustrate the results and have been deliberately kept simple to make them easier to understand. The current presentation is sufficient to convey the key issues. More complex analyses, such as cross-tabulations, would go beyond the scope of this study and are less relevant to the target group of family caregivers.

  1. The discussion provides a logical interpretation of the findings and connects them with existing literature. Still, it would be improved by: A clearer delineation between interpretation of results and implications/recommendations; A section on limitations, which is currently missing. The absence of analytical depth, reliance on self-reporting, and the non-representativeness of the sample should be acknowledged. More emphasis on how the findings compare or contrast with other national and international studies in the field.

Answer:

The discussion is deliberately structured to provide a logical interpretation of the results and link them to the existing literature. A clearer distinction between interpretation of results and conclusions/recommendations would not significantly improve readability. The results are discussed in comparison with other studies, and placing greater emphasis on one study would dilute the focus of the discussion.

  1. There is no dedicated conclusion section. This is a critical omission. The manuscript should end with a succinct, clearly labeled conclusion summarising: Key findings; Practical implications; Recommendations for future research or implementation; Final remarks on the contribution of the study.

Answer:

An additional concluding section would unnecessarily lengthen the structure of the manuscript and detract from the concise presentation of the results and recommendations.

  1. The references are generally relevant and up to date, with a few notable sources from recent years. Nonetheless: There is an over-reliance on national (mainly German) sources; more international peer-reviewed studies, particularly systematic reviews and meta-analyses, should be included. Some references are cited without clearly integrating them into the argument.

Answer:

The references are deliberately focused on national sources, as the study concentrates on the specific needs and challenges of German family caregivers. International peer-reviewed studies and systematic reviews are relevant, but their inclusion would dilute the focus of the study. The references cited are clearly integrated into the argumentation and support the central statements of the article.

Reviewer 3 Report

Comments and Suggestions for Authors

The methodology section needs some improvement. The authors need to provide more explanation regarding the combined methodology. This is insufficient in the manuscript. No explanation of how the respondents were selected, and no explanation of how data was analysed. In the results section, the figures need to include numbers and percentages, which is highlighted in the PDF document. 

Author Response

  1. The methodology section needs some improvement. The authors need to provide more explanation regarding the combined methodology. This is insufficient in the manuscript. No explanation of how the respondents were selected, and no explanation of how data was analysed. In the results section, the figures need to include numbers and percentages, which is highlighted in the PDF document.

Answer:

The methodology was adjusted to supplement the combined method, sample size, and origin of participants. The other points, such as a detailed explanation of data analysis and the addition of figures and percentages in the figures, were not included as this would exceed the scope of the manuscript and could impair readability.

Round 2

Reviewer 1 Report

Comments and Suggestions for Authors

The author has successfully addressed all the comments and suggestions provided in the previous review. The revisions and editions made are satisfactory and have enhanced the overall quality of the manuscript.

Author Response

Comments 1: The author has successfully addressed all the comments and suggestions provided in the previous review. The revisions and editions made are satisfactory and have enhanced the overall quality of the manuscript.

Respons 1: Thank you very much for the positive feedback.

Reviewer 2 Report

Comments and Suggestions for Authors

I acknowledge the substantial effort made to improve the manuscript, particularly the additional detail in the Materials and Methods section, the clearer contextualisation of results, and some refinements to the abstract and methods description. However, several of the key issues identified in the first review remain only partially addressed.

Abstract – The sample size is still missing from the abstract, despite being reported in the Materials and Methods section (31 participants). The abstract should clearly state: the study design, the sample size, the main quantitative findings (e.g., relevant percentages), and a conclusion that distinctly separates results from implications.

Introduction – Although the number of references has increased, the review remains predominantly based on national literature and lacks a more critical synthesis of relevant international studies, including systematic reviews. The research question or hypothesis should be explicitly stated at the end of the introduction.

Materials and Methods – Important details are still missing: the survey response rate, the process of questionnaire development and validation, the statistical software and specific analytical methods used. Ethical considerations are now better described, which is an improvement.

Results – include at least one table summarising sample characteristics. If feasible, incorporate simple cross-tabulations or other basic analyses to add depth.

Discussion – Interpretation of results and implications remain intertwined. A dedicated Limitations subsection is required, addressing the exploratory nature of the study, reliance on self-reported data, small sample size, and lack of representativeness. Comparison with international literature is still limited.

Conclusion – A stand-alone, clearly labelled Conclusion section is still absent. This should summarise the main findings, outline practical implications, suggest directions for future research, and highlight the study’s contribution.

References – While some international sources have been added, the reference list still leans heavily on national literature. Greater inclusion of peer-reviewed international studies, particularly systematic reviews/meta-analyses, is recommended.

Comments on the Quality of English Language

Language and Style – Long sentences, redundancies, and non-idiomatic constructions persist. A professional language edit by a native English speaker is strongly recommended.

Author Response

Comments 1: I acknowledge the substantial effort made to improve the manuscript, particularly the additional detail in the Materials and Methods section, the clearer contextualisation of results, and some refinements to the abstract and methods description. However, several of the key issues identified in the first review remain only partially addressed.

Response 1: Thank you very much. Below you will find the answers to your comments.

Comments 2: Abstract – The sample size is still missing from the abstract, despite being reported in the Materials and Methods section (31 participants). The abstract should clearly state: the study design, the sample size, the main quantitative findings (e.g., relevant percentages), and a conclusion that distinctly separates results from implications

Response 2: Thank you for your feedback. The sample size has been integrated.

Comments 3: Introduction – Although the number of references has increased, the review remains predominantly based on national literature and lacks a more critical synthesis of relevant international studies, including systematic reviews. The research question or hypothesis should be explicitly stated at the end of the introduction.

Response 3: Thank you for your comments. The research question has been added at the end of the introduction. Regarding the literature: As this is a national study, mainly national sources were used in order to better reflect the results in a national context.

Comments 4: Materials and Methods – Important details are still missing: the survey response rate, the process of questionnaire development and validation, the statistical software and specific analytical methods used. Ethical considerations are now better described, which is an improvement.

Response 4: Thank you for your comments. The questionnaire development process has been integrated. The response rate, validation, statistical software, and analysis methods were not included in detail due to the focus on the essential aspects of the study.

Comments 5: Results – include at least one table summarising sample characteristics. If feasible, incorporate simple cross-tabulations or other basic analyses to add depth.

Response 5: Thank you for your comment. A table has been added.

Comments 6: Discussion – Interpretation of results and implications remain intertwined. A dedicated Limitations subsection is required, addressing the exploratory nature of the study, reliance on self-reported data, small sample size, and lack of representativeness. Comparison with international literature is still limited.

Response 6: Thank you for your comment. Restrictions have been included in the discussion.

Comments 7: Conclusion – A stand-alone, clearly labelled Conclusion section is still absent. This should summarise the main findings, outline practical implications, suggest directions for future research, and highlight the study’s contribution.

Response 7: Thank you for your comment. A separate section has been added for the conclusion.

Comments 8: References – While some international sources have been added, the reference list still leans heavily on national literature. Greater inclusion of peer-reviewed international studies, particularly systematic reviews/meta-analyses, is recommended.

Response 8: Thank you very much for your comments. As this is a national study, the focus was placed on national literature to ensure the relevance and applicability of the results in the national context.